# Risk Assessment of An Earthquake-Collapse-Landslide Disaster Chain by Bayesian Network and Newmark Models

**DOI:** 10.3390/ijerph16183330

**Published:** 2019-09-10

**Authors:** Lina Han, Qing Ma, Feng Zhang, Yichen Zhang, Jiquan Zhang, Yongbin Bao, Jing Zhao

**Affiliations:** 1School of Environment, Northeast Normal University, Changchun 130024, Chinazhaoj870@nenu.edu.cn (J.Z.); 2State Environmental Protection Key Laboratory of Wetland Ecology and Vegetation Restoration, Northeast Normal University, Changchun 130117, China; 3Key Laboratory for Vegetation Ecology, Ministry of Education, Changchun 130117, China; 4College of Resources and Environment, Jilin Agricultural University, Changchun 130118, China; 5Jilin Institute of Geological Environment Monitoring, Changchun 130061, China

**Keywords:** risk assessment, earthquake disaster chain, Bayesian Network model, Newmark model, Changbai Mountain volcano

## Abstract

Severe natural disasters and related secondary disasters are a huge menace to society. Currently, it is difficult to identify risk formation mechanisms and quantitatively evaluate the risks associated with disaster chains; thus, there is a need to further develop relevant risk assessment methods. In this research, we propose an earthquake disaster chain risk evaluation method that couples Bayesian network and Newmark models that are based on natural hazard risk formation theory with the aim of identifying the influence of earthquake disaster chains. This new method effectively considers two risk elements: hazard and vulnerability, and hazard analysis, which includes chain probability analysis and hazard intensity analysis. The chain probability of adjacent disasters was obtained from the Bayesian network model, and the permanent displacement that was applied to represent the potential hazard intensity was calculated by the Newmark model. To validate the method, the Changbai Mountain volcano earthquake–collapse–landslide disaster chain was selected as a case study. The risk assessment results showed that the high-and medium-risk zones were predominantly located within a 10 km radius of Tianchi, and that other regions within the study area were mainly associated with very low-to low-risk values. The verified results of the reported method showed that the area of the receiver operating characteristic (ROC) curve was 0.817, which indicates that the method is very effective for earthquake disaster chain risk recognition and assessment.

## 1. Introduction

In recent years, countries and regions affected by major natural disasters have been increasingly reported, the occurrences of which have led to a great number of casualties and serious economic losses. In 2008, a devastating earthquake of magnitude 8.0 occurred in Wenchuan, Sichuan Province, China. This triggered a series of disaster chains such as an earthquake–landslides–reservoir disaster chain and an earthquake–landslide–debris flow disaster chain [1,2]. The total number of casualties was 69,225, and approximately one quarter of those casualties related to secondary landslides caused by the earthquake [3]. In 2013, a disastrous earthquake of magnitude 7.0 took place in Lushan, Sichuan Province, China. The event induced many landslides and caused a significant number of casualties [4]. In 2017, an earthquake of magnitude 7.0 occurred in Sichuan Province, China. The earthquake triggered numerous geological disasters such as landslides and debris flows, which severely destroyed the natural landscape of the Sparkling Lake and the Panda Lake Waterfall [5,6]. Frequent disaster events have shown that for most major disasters, secondary disasters are always induced; ultimately combining as a considerable destructive power. The phenomenon of secondary disasters being caused by some kind of primary disaster is considered to be a disaster chain [7]. More importantly, the casualties and property damage related to disaster chains are deemed to be greater than those resulting from the primary source disasters themselves [8]. Therefore, disaster chain risk assessment has become one of the urgent core issues to be addressed in current international research.

There are currently three disaster chain risk assessment methods. The first is the probabilistic analysis method based on data. Examples include that presented by Gupta [9], which involves a probabilistic risk assessment method for structural systems based on the Bayesian network framework under multiple hazards, and that presented by Korswagen [10], who explored a probabilistic risk assessment framework toward the structural damage of masonry housing induced by earthquakes and secondary floods. The second method is based on complex networks, for example, presented by Zheng [11], who adopted a complex network model to build a natural hazard network to research disaster chain mechanisms. Liu [12] also put forward a disaster chain risk evaluation model on the basis of complex networks. The third method is based on remote sensing, such as that undertaken by Meena [13], who applied remote sensing and geographic information systems (GIS) to map earthquake-induced landslide susceptibility zones. A further example is that of Sharma [14], who used space remote sensing techniques to analyze time and space satellite images of landslide induced by earthquake. Therefore, earthquake disaster chain risk evaluation, on the basis of natural hazard risk formation theory, is currently rare.

Many different assessment models can be used for disaster hazard and risk research. Artificial neural network models, logistic regression models, and Bayesian network (BN) models are often used for probability analysis of earthquake-induced secondary disasters [15,16,17]. Finite element modeling, pseudo-static analysis, and the Newmark model are frequently applied to depict the slope behavior of disaster events during earthquakes [18,19,20]. Among them, a BN model provides a useful way to deal with complicated problems as it can combine probabilistic inference methods with a graphical representation that reveals causal relationships between different network nodes, and thus offers a network structure for handling uncertainty and complexity [21,22,23]. Song [17] proposed a hybrid method based on a BN to assess the susceptibility of an earthquake-induced landslide. Vogel [24] applied the various learning algorithm of the BN to assess the hazards of earthquakes, floods, and landslides. Ozdemir [25] analyzed the relationships between landslide distributions with 19 landslide related parameters by a Bayesian model. The Newmark model has been used more widely than pseudo-static analysis and finite element modeling in specific slope analyses [26,27], and earthquake-induced secondary disaster analyses [28,29] are considered to yield much more helpful information. Caccavale [30] proposed an integrated approach to assess earthquake-induced landslide hazards based on the Newmark method and frequency–magnitude curves. Liu [31] put forward Newmark’s sliding rigid-block model to calculate cumulative displacements and to identify potentially unstable areas. Chousianitis [32] developed an empirical estimator of co-seismic landslide displacements based on the Newmark model to assess the hazards related to earthquake-induced landslides. Del Gaudio [33] used the Newmark model to analyze seismic hazards in landslide-prone regions.

The author’s previous work was mainly to construct the hazard assessment model of earthquake–collapse–landslide–debris flow disaster chain based on a BN from the perspective of every single disaster according to the hazard formation mechanism, which included chain probability and hazard intensity, and were obtained from inference of BN [34]. There are still some deficiencies in the previous works, first, the risk of the earthquake disaster chain was not considered, and the hazard discussed in the previous works was one factor in risk assessment elements. Second, BN cannot adequately reflect the nature of the impact of earthquake on secondary disasters, and cannot well calculate the hazard intensity of earthquake disaster. Finally, the hazard of earthquake disaster chain was analyzed from a single disaster perspective. Based on the above deficiencies, this study put forward a volcanic earthquake–collapse–landslide disaster chain risk evaluation model on the basis of a BN model and Newmark model. The risk evaluation model was constructed according to natural disaster risk formation theory from the perspective of the overall disaster chain. The BN can be used to analyze chain probability between the adjacent disaster events, and the hazard intensity of earthquake on secondary disasters can be analyzed according to the permanent displacement calculated in the Newmark model as the permanent displacement obtained from the Newmark model can better describe the impacts of earthquakes on secondary disasters. To validate the effectiveness of the assessment method, the earthquake–collapse–landslide disaster chain induced by the Changbai Mountain volcano eruption was used as a case study. The earthquake disaster chain risk assessment model and case study presented in the study aim to offer a framework and tool for seismic disaster chain risk identification on the basis of natural hazard risk theory, which considers the chain probability of the disaster environment, hazard intensity of the hazard factor, and the vulnerability of the disaster body.

## 2. Methods and Materials

### 2.1. The Study Area

The Changbai Mountain is a worldly renowned scenic area situated in the southeast region of Jilin Province, China, and has a total area of ~3278 km^2^. Due to the local fracture movement, volcanic seismic activity has shown a clear increased trend since 2002. More than 200 earthquakes with magnitudes >1 have occurred every year since 2002. The Changbai Mountain volcanic structure is very developed, and Tianchi is considered to be the center with 12 fault zones surrounding it, as illustrated in Figure 1. This includes eight radioactive fault zones and four ring-shaped fault zones. A nearby area of two radioactive fault zones (F6 and F7) with a north–south U-shaped canyon is the most developed and intensive rock fracture section in the study area [35]. In these regions, the topography and geomorphology are highly complex and are conductive to the frequent occurrence of geological disasters with strong, sudden, and destructive characteristics. According to the Changbai Mountain geological disaster investigation and zoning report, there were 108 collapses and landslide disasters between 2000 and 2017. As the Changbai Mountain is a renowned tourist area and national nature reserve, it attracts numerous tourists. More than 20,000 people (local and tourists) have been affected by secondary disasters in the area.

Most studies [36,37,38,39] have shown that: (1) the earthquakes were triggered by the local fracture movement in the volcanic zone; (2) the structure of the rock-soil was damaged; (3) the originally weak structural planes and joint fissures in the stratum were displaced; (4) the bedrock fissure network was penetrated, expanded, further evolved, and accumulated into loose debris under specific hydrological conditions; and (5) subsequent to steps 1–4, collapse and landslide occurred. As a result, an earthquake–collapse–landslide disaster chain was formed.

### 2.2. Bayesian Network Model

BNs are interoperable, speedy, and effective modeling tools that are used in risk and parameter assessment in complicated systems under major indeterminacy [40]. The qualitative component of a BN is a directed acyclic graph (DAG) that delimits the factorization of joint probability distribution in the specific system, which is expressed by the network nodes and directed links [41]. The quantitative part of a BN is the network parameters, which specify the conditional probability of every node under the conditions of its father nodes [40]. The dependencies between parent nodes with child nodes are quantitatively represented by a conditional probability table (CPT), which can be obtained from expert decision [42,43] or from training based on historical data [44,45]. Each node of a BN model is divided into a finite suit of status values represented by qualitative or quantitative values. Therefore, BN models work out discrete probabilities in a general way. In a BN analysis, the posterior probability is determined by the relationship in Equation (1), under the conditions of given evidence or data *B* [46]:(1)$$P\left(A|B\right)=\frac{P\left(B|A\right)P\left(A\right)}{P\left(B\right)}$$
where *P(A|B)* is the posterior probability of *A* on the basis of data or evidence *B*; *P(B|A)* is the prior probability of *B* given *A*; and *P(A)*, *P(B)* are the marginal probabilities.

In this study, a BN model was represented by a graphical network that showed the causality relationships between the diverse disaster events and related factors. Seven major parameters (earthquake intensity, elevation, slope, slope aspect, lithology, distances to rivers, and precipitation) were selected, according to geo-environmental conditions, and the disaster chain mechanism indicated by earthquake-induced secondary disaster research [1,46,47]. Therefore, the earthquake, collapse, landslide disaster events, and related factors were considered as network nodes, and the links of these nodes were considered as edges. The BN structure map was constructed and is shown in Figure 2. A total of 300 disaster points selected for the Changbai Mountain region (108) and Jilin province (192) were divided into training and test subsets. Among them, 50 disaster points obtained from the Changbai Mountain were used for the validation of the results and the rest were used for parameter learning.

As a result, it was possible to update the probability in the BN when new evidence or data became available. When the status of a node was changed, the information would propagate via the whole network, and the probability of every related node would change. The entire area of study was divided into 1 km × 1 km grid cells, the total number of grid cells was 5922, and all attribute values (elevation, slope, aspect, lithology, distances to rivers, and precipitation) of each cell’s center-point were extracted by the spatial analyst tools in the ArcGIS software. The obtained grid cell data were converted into a case file format and input to the BN model in the Netica software so that the posterior probability of the three disaster events could be obtained from the inference of the BN model. The data obtained from the BN model were input into Excel software and merged in the ArcGIS software, which were then converted into raster data before being created as output.

### 2.3. Newmark Model

The basic principle of the Newmark model is to measure cumulative displacement induced by the seismic acceleration. The specific value of the cumulative displacement is calculated by quadratic integration of the seismic acceleration that exceeds the critical acceleration. Jibson [48] used 2270 strong vibration records in 30 global earthquakes to establish an empirical model, as expressed by Equation (2):(2)$$\mathrm{lg}(D_{N})=2.401\mathrm{lg}\left(I_{a}\right)-3.481\mathrm{lg}\left(a_{c}\right)-3.23\pm 0.656$$
where *D_N_* is the cumulative displacement (cm); *Ia* is the Arias intensity (m/s); *a_c_* is the critical acceleration (m/s^2^); ±0.656 represents the fluctuation of the cumulative displacement value; and lg is the Briggs logarithm (base 10).

Arias intensity is an index that records the amplitude, frequency, and duration time of an earthquake. Thus, the Arias intensity is deemed superior or more suitable than other earthquake arguments for use in collapse and landslide assessments related to earthquakes. As the acquisition of this parameter is based on the instrument’s record, the Arias intensity for the entire study area must be indirectly obtained by the available seismic parameters. Wilson [49] provided an empirical formula, as shown by Equation (3), for determining an *Ia* value from the peak ground acceleration (PGA) based on 43 seismic records:(3)$$Ia=0.9t_{d}{\left(a_{max}\right)}^{2},$$
where *a_max_* is the peak acceleration, and *t_d_* is the Dobry duration time, which is expressed by Equation (4):(4)$$\mathrm{lg}(t_{d})=0.432M-1.83$$
where *M* is the earthquake magnitude according to the Richter scale.

The seismic intensity corresponding to different epicenter distances, *r* (km), can be obtained according to the earthquake intensity attenuation formula in eastern China [50,51], as denoted by Equation (5):(5)I=4.493+1.454M−1.792ln(r+16),
where ln is the natural logarithm (base e).

The empirical formula for seismic intensity and peak ground acceleration in mainland China can be acquired by seismic intensity tables [52] and Equation (6):(6)$$I=3.322\mathrm{lg}\left(a_{max}\right)+0.033$$

Based on Equation (6), the Arias intensity values at different epicenter distances could be obtained under conditions of known magnitude for the entire study area.

The critical acceleration is an acceleration value when the slope’s sliding force under the action of seismic load is equal to the slope’s anti-sliding force; its value reflects the ability of the slope to resist earthquake damage and is affected by the material composition and the slope [30,53]. When the accelerated velocity of an earthquake surpasses the critical acceleration of an unstable slope under the conditions of an earthquake, the slope may slip along the failure plane, and permanent displacement will amass in the downward orientation [54]. The critical acceleration can be obtained from Equation (7) [55]:(7)$$a_{c}=\left(F_{s}-1\right)g\sin\alpha ,$$
where *α* is the gradient of the slope; *g* is the gravitational acceleration; and *Fs* is the static safety factor, which can be represented by Equation (8) [56,57]:(8)$$F_{S}=\frac{C^{\prime }}{\gamma t\sin\alpha }+\frac{\tan\varphi {}^{\prime }}{\tan\alpha }-\frac{m\gamma_{w}\tan\varphi^{\prime }}{\gamma \tan\alpha },$$
where *C’* is the efficient cohesion (MPa); *φ’* is the internal friction angle (°); *γ* is weight of rock-soil body (N/m^3^); *γ_W_* is the weight of water (N/m^3^); *t* is the sliding plane thickness (m); and *m* is the thickness ratio of the failure plane under water.

### 2.4. Earthquake Disaster Chain Risk Assessment Model

According to the natural hazard risk formation mechanism, in a certain region, a natural hazard risk is developed from the comprehensive interaction of four risk elements: hazard factors, the exposure of a disaster body, the vulnerability of a disaster body, and emergency response and recovery capability [58]. In this study, hazard and vulnerability were considered in the earthquake disaster chain risk evaluation model. We defined hazard as the influence of the activity scale, and the frequency of the hazard factors on the disaster body. Disaster chain hazard analyses include a susceptibility analysis of the disaster chain (chain probability) and a hazard intensity analysis of the disaster chain (the sum of the hazard intensities of each disaster event). Hazard can be simplistically expressed by Equation (9):(9)H=P×I
where H is the hazard value of a disaster chain; P is the chain probability of a disaster chain; and I is the hazard intensity of a disaster chain. The chain probability is the possibility that secondary disasters will be induced by a primary disaster, and this possibility is related to the hazard intensity of the primary disaster and the sensitivity of the disaster-prone environment. The hazard intensity is the degree of destruction for the disaster-prone environment and secondary disasters. The chain probability was obtained from the BN inference in Netica software. Hazard intensity was expressed in terms of the cumulative displacement that was calculated by the Newmark model.

In this work, vulnerability refers to the degree to which the disaster body may suffer losses under the intensity of certain hazard factors. When major geological disasters occur, they often directly threaten the lives of people in the affected areas, and the size of the population directly influences the number of casualties. In developed areas of society and the economy, great losses often result, but disaster resistance can also be relatively strong. The Changbai Mountain is a national nature reserve; major geological disasters have often destroyed the vegetation and caused direct or indirect property losses. For these reasons, our vulnerability analysis included the density of the population, per capita gross domestic product (GDP), and a normalized difference vegetation index (NDVI) [59]. The procedures of the integrated model for earthquake disaster chain risk identification are displayed in Figure 3. Overall, the disaster chain risk evaluation model can be denoted by Equation (10):(10)R=H×V,
where R is the risk value of a disaster chain; H is the hazard of a disaster chain; and V is the vulnerability of a disaster body.

### 2.5. Data and Model Parameters

The dataset required for the BN and Newmark models includes: (1) an earthquake shaking parameter, that is, Arias intensity calculated by empirical formula; (2) a lithology map to obtain the static safety factor and critical acceleration; (3) a digital elevation model (DEM) to obtain the slope; and (4) a rainfall contour map. All parameters were calculated based on raster grids.

The Arias intensity of the whole study area must be obtained from the seismic parameters indirectly due to a lack of data. In 1991, the Mount Pinatubo volcano erupted, and the earthquake magnitude was 5.6, which was the largest volcano associated earthquake magnitude recorded globally to date [60]. Based on this, a magnitude 6.0 earthquake can be predicted in relation to a hypothetical eruption of the Changbai Mountain volcano. According to Equations (3)–(6), it is possible to obtain the seismic intensity and Arias intensity from ArcGIS based on the zone map of the study area, as shown inFigure 4a and Figure 5a, respectively.

The lithology map for the area was digitized from 1:200,000 geological maps prepared by the Jilin Institute of Geological Environment Monitoring (JIGEM). According to GB50218T-2014 (China), the rock-soil masses can be classified into four categories (Figure 4b). The studied area is mainly characterized by hard rock, which makes up ~75% of the influenced region. The soft rock mainly includes alkaline trachyte and rough breccia that is distributed around the Tianchi. The parameter values (i.e., efficient cohesion, internal friction angle, and rock-soil body weight) of these rock-soil masses are listed in Table 1. According to Equations (7) and (8), the static safety factor and critical acceleration can be obtained from ArcGIS, as shown in Figure 5b,c. According to Equation (2), the cumulative displacement map can also be obtained (Figure 5d).

A DEM (30 m × 30 m) was extracted from a 1:50,000 topographic maps prepared by the JIGEM. Slope aspect, slope, and elevation were obtained from the DEM through ArcGIS spatial analysis tools. The elevation of the whole studied region ranged from 609 m to 2681 m (Figure 2). The slope ranged from 0° to 61.23° (Figure 4c), and the slope aspect is shown in Figure 4d. The drainage lines were digitized from a drainage line distribution dataset with a 1:250,000 scale, and the map of the distance to the river is shown in Figure 4e. The precipitation map (Figure 4f) can be acquired from the Jilin Institute of Geological Environment Monitoring JIGEM.

## 3. Results and Discussions

### 3.1. Earthquake Disaster Chain Hazard Analysis

#### 3.1.1. Chain Probability Analysis

The chain probability is the possibility that secondary disasters will be induced by a primary disaster, and this possibility is the conditional probability of a secondary disaster event under the influence of a primary disaster event. Moreover, under the influence of a primary disaster with different hazard intensity, the loss levels of a secondary disaster event are also different [12]. Therefore, it is necessary to obtain the probability of the different secondary disaster loss levels under the influence of different hazard intensity levels of the primary disaster. This probabilistic reasoning process can be implemented by the BN model in Netica software. In addition, the chain probability of the earthquake–collapse–landslide disaster chain is the product of all of the occurrence probabilities of disaster events, as illustrated in Figure 6a. From the visualization analysis of this map, zones of high chain probability were predominantly located within the central area of the studied area, with a distribution area, accounting for approximately one third of the study area. Nevertheless, the northern and southern regions primarily showed a low chain probability, with the distribution area in this case comprising approximately half of the study area.

#### 3.1.2. Hazard Intensity Analysis

The hazard intensity and occurrence probability are always inseparable. Generally speaking, if the hazard intensity is greater, the occurrence probability is smaller. The hazard intensity is expressed by the probability, or the transcendental probability, in some studies. The transcendental probability is the probability that an engineering site may encounter a value greater than or equal to either a given seismic intensity or ground motion parameter within a certain period of time [61,62]. The cumulative displacement of collapse and landslides induced by earthquake is shown in Figure 5d. The higher permanent displacement values reflect the higher hazard intensity. Therefore, the hazard intensity of an earthquake disaster can be represented by the cumulative displacement calculated by the Newmark model. In this study, the related parameters of the model were dealt with by ArcGIS software and calculated by a Raster calculator. The hazard intensity of collapse and landslide was expressed by the hazard areas obtained from the inference of the BN model. The data obtained from the Newmark and BN models were normalized and the hazard intensity of the earthquake disaster chain was the sum of every disaster event’s hazard intensity (Figure 6b). From visualized analysis of the obtained hazard intensity map, the high hazard intensity regions were observed to be situated in 10 km semi diameters focused on the Tianchi center. It is evident that the spatial trend of the earthquake disaster chain hazard zone follows the distribution of elevation and slope. Zones of very high hazard were predominantly located within a 15-km radius of Tianchi in the previous work [34]. The main reason for the above differences is that the hazard intensity of the earthquake–collapse–landslide–debris flow disaster chain in previous work comes from the inference of BN; in this study, the hazard intensity analysis of volcanic earthquake–collapse–landslide disaster chain were obtained from the cumulative displacement calculated by the Newmark model. From the comparison of the results in the two works, it can be seen that the range of hazard intensity obtained in this study was smaller, while the number of disaster points situated in the high hazard zones was higher, so the hazard assessment results in this study were more accurate. This result also shows that the application of the Newmark model is correct.

### 3.2. Earthquake Disaster Chain Vulnerability Analysis

The vulnerability analysis of a disaster body in a disaster chain is more complicated than the hazard intensity and probability analyses. This is partly because the vulnerability of a disaster body can change due to repeated exposure to the same (or new) hazard factor. Previous vulnerability studies have been based on vulnerability curves, which did not consider the vulnerability changes of a disaster body in the disaster chain [63,64,65]. There are no related records of casualties or property loss caused by volcanoes and earthquakes that occurred several hundred years ago. Therefore, the population density, per capita GDP, and NDVI (normalized difference vegetation index) are mainly considered for the vulnerability analysis of an earthquake disaster chain. The normalized visualization result of this is shown in Figure 6c, and shows that the high vulnerability zones were situated in the Chi west region, while the Chi south region mainly contained very low vulnerability values.

### 3.3. Earthquake Disaster Chain Risk Analysis

With regard to the earthquake disaster chain risk evaluation method presented in this research, the obtained risk map is shown in Figure 6d. This divided the earthquake–collapse–landslide disaster chain risk into four categories through an equal interval classification method according to risk value (very low: 0–0.2; low: 0.2–0.4; medium: 0.4–0.6; high: >0.6), which ranged from unsusceptible to susceptible. Furthermore, the 50 disaster point locations of validation (Figure 6d) were mainly situated in high risk zones. Only a few disaster points were located in low-risk areas, which indicates that the results of the disaster chain risk assessment were good. From the visualized analysis of the disaster chain risk zone, the high- and medium-risk zones were predominantly located within a 10 km radius of Tianchi, whereas the other regions of the study area principally contained very low-or low-risk values. Due to the elevation, the slope was significantly different in some areas, thereby resulting in a relatively obvious gradient between the Tianchi surroundings and other regions of the study area. More importantly, the spatial trend of the earthquake disaster chain risk zone followed the zones of hazard intensity in high zones. Therefore, in the disaster chain risk’s formation process, the favorable factor is the hazard intensity of the hazard factor, with the hazard itself being a major factor in the four elements of disaster risk.

### 3.4. Validation of the Risk Assessment Model

The validation of an earthquake disaster chain risk evaluation model aimed to estimate the relative distribution between the disaster chain risk zones and number of disaster events that have occurred. The results shown in Figure 7 indicate an increase in the disaster events number ratio when moving from low-risk classes to high-risk classes. The quantitative validation results showed that 31 out of 50 disaster points were situated in high-risk areas, thus comprising 62% of the study area. Ten disaster points were situated in regions with medium-risk levels, thus comprising 20% of the study area. Therefore, 82% of all disaster points occurred in the medium-to high-risk region, that is to say, the disaster chain risk assessment model is very applicable for use in susceptibility assessments of earthquake disaster chains.

The earthquake disaster chain risk assessment model was also verified by applying the receiver operating characteristic (ROC) method [66]. This allowed us to evaluate the model’s predictive capability for a particular probability threshold, which may be chosen to classify either a disaster point or a non-disaster point. The prediction accuracy for the risk assessment method was evaluated quantitatively by the area under the curve (AUC), with values ranging from 0.5 (no accurately predictive ability) to 1.0 (perfect accuracy). In particular, AUC values >0.7 can be considered as indicating an acceptable predictive value [67]. The ROC curve of the earthquake disaster chain risk evaluation results was constructed by the relationships of the risk zoning map (high-and medium-risk zones were considered as occurrence, and low-and very low-risk zones were considered as non-occurrence). This and the disaster point locations (that occurred in the past) from SPSS software is shown in Figure 8. The T_0_^a^ varied from 0 to 5.9 in step of 0.1 in order to more precisely draw the ROC curve, the value for the “best threshold” was 2.9. The “best threshold” was the top left point (red), the point on the ROC curve at a minimum distance from (0, 1) was chosen, and its numbers of TP, TN, FP, and FN (True Positives, True Negatives, False Positives, and False Negatives) are shown in Table 2. The AUC value from Figure 8 was 0.817, which suggests that the accuracy of the disaster chain risk evaluation results was sufficiently high.

## 4. Conclusions

Most major natural hazards can trigger a series of catastrophically secondary disasters, either simultaneously or sequentially, due to the disaster chain characteristics of temporal inducibility and spatial sprawl. The risk assessment of disaster chains is more complicated than that of individual disasters because the primary disaster can trigger a series of secondary disasters, and it is very difficult to recognize the interactions and chain mechanism involved. Despite more attention having been paid to the relationships between different disaster events, there is currently still no uniform conceptual model that can be used to evaluate the earthquake disaster chain risk. Compared with individual disasters, there are more hazard factors in a disaster chain. Furthermore, the vulnerability of a disaster body is changed by multi-hazard factors, where not only are there more disaster events, but the spatial scope of damage is also greater in a multi-hazard environment. According to major disasters of the past, there is therefore a real need for a new method for earthquake disaster chain risk evaluation. In this research, a new earthquake disaster chain risk evaluation conceptual model that coupled a BN and Newmark models was proposed on the basis of natural hazard risk formation theory. The new method combined these models on the basis of Excel and ArcGIS software, which is useful for the identification of quantitative risk parameters including chain probability, hazard intensity, and vulnerability. The chain probability of an earthquake disaster chain was obtained from the BN model, the hazard intensity of the earthquake disaster chain was calculated by the Newmark model. The joint method successfully highlights the comprehensive recognition of the disaster chain formation mechanism and quantitatively assesses disaster chain risk.

Using this joint method, the risk assessment results for the Changbai Mountain volcano earthquake disaster chain were obtained. The risk map showed that the high- and medium-risk zones were predominantly located within a 10 km radius of Tianchi, whereas the other regions of the study area primarily contained very low- or low-risk values. The verification results showed that the area under the ROC curve was 0.817, a value reasonably in agreement with both the AUROC and with the value obtained in the validation procedure on the test subset, thus suggesting that the simulated results based on this new method were coincident with disaster events of the past. The earthquake disaster chain risk assessment model proposed in this study provides a reference for the prevention and mitigation of disaster chains in mountainous area.

Although the joint method presented good performance for the earthquake disaster chain risk evaluation in the Changbai Mountain region, obstacles still exist for the assessment of disaster chain risk. The chain formation process from the primary to the secondary disasters is complicated, particularly in the vulnerability changes of the disaster body due to being repeatedly damaged by the same (or new) hazard factors. Therefore, a lack of consideration of the vulnerability changes for earthquake disaster chain risk assessment is a significant defect, and represents an issue for further study.

## Figures and Tables

**Figure 1 ijerph-16-03330-f001:**
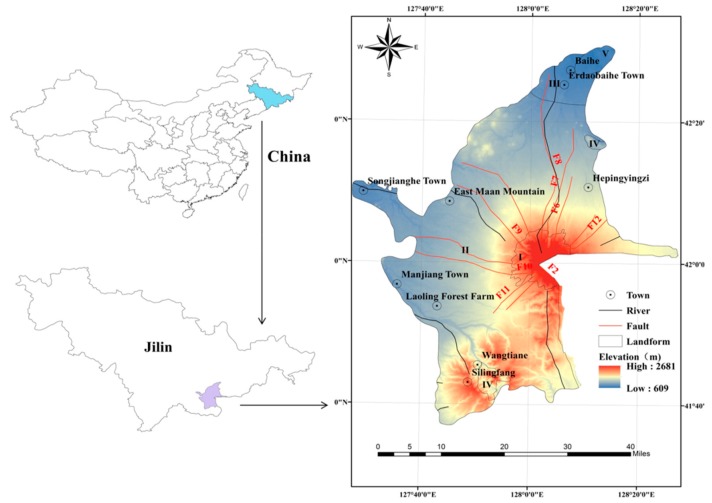
Location and elevation map of the study area (I: Volcanic cone; II: Lava Taiyuan; III: Melting platform; IV: Tectonic denudation of middle and low mountains; V: Erosional valley).

**Figure 2 ijerph-16-03330-f002:**
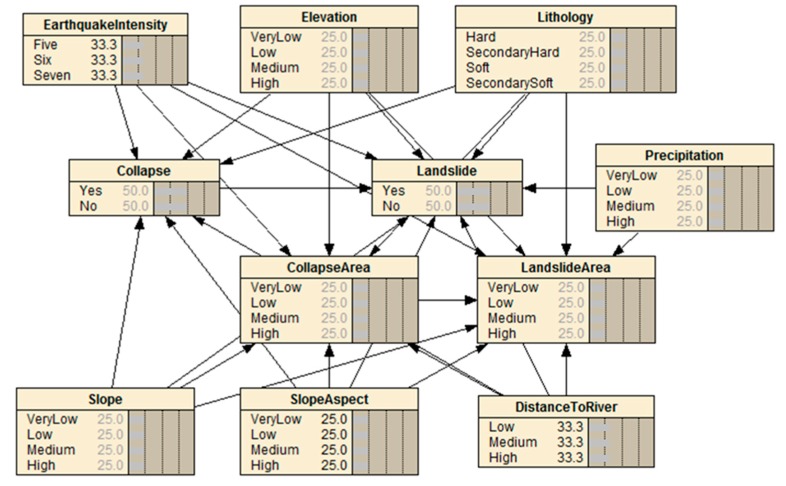
The Bayesian network structure of the Changbai Mountain earthquake disaster chain.

**Figure 3 ijerph-16-03330-f003:**
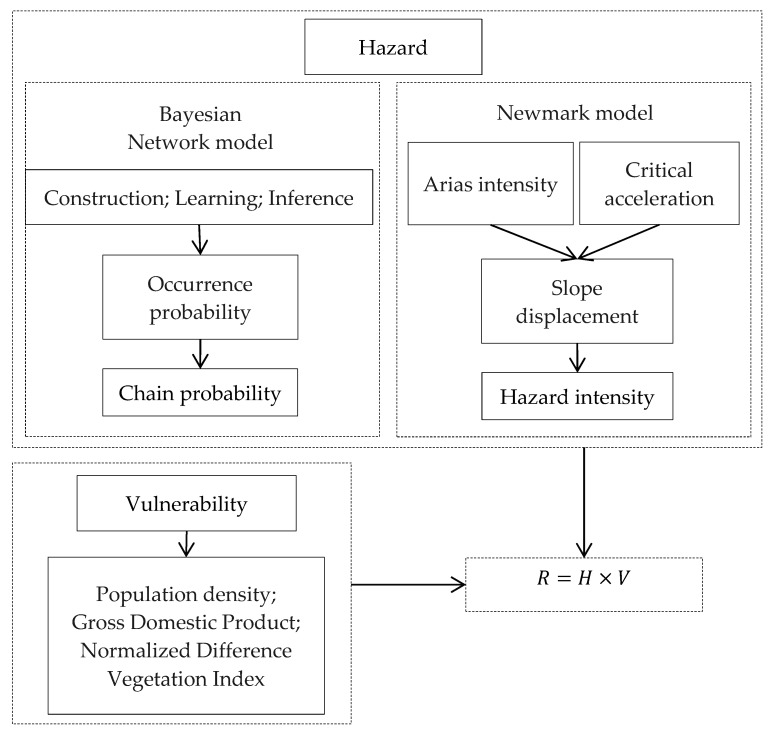
The risk identification steps of the earthquake disaster chain.

**Figure 4 ijerph-16-03330-f004:**
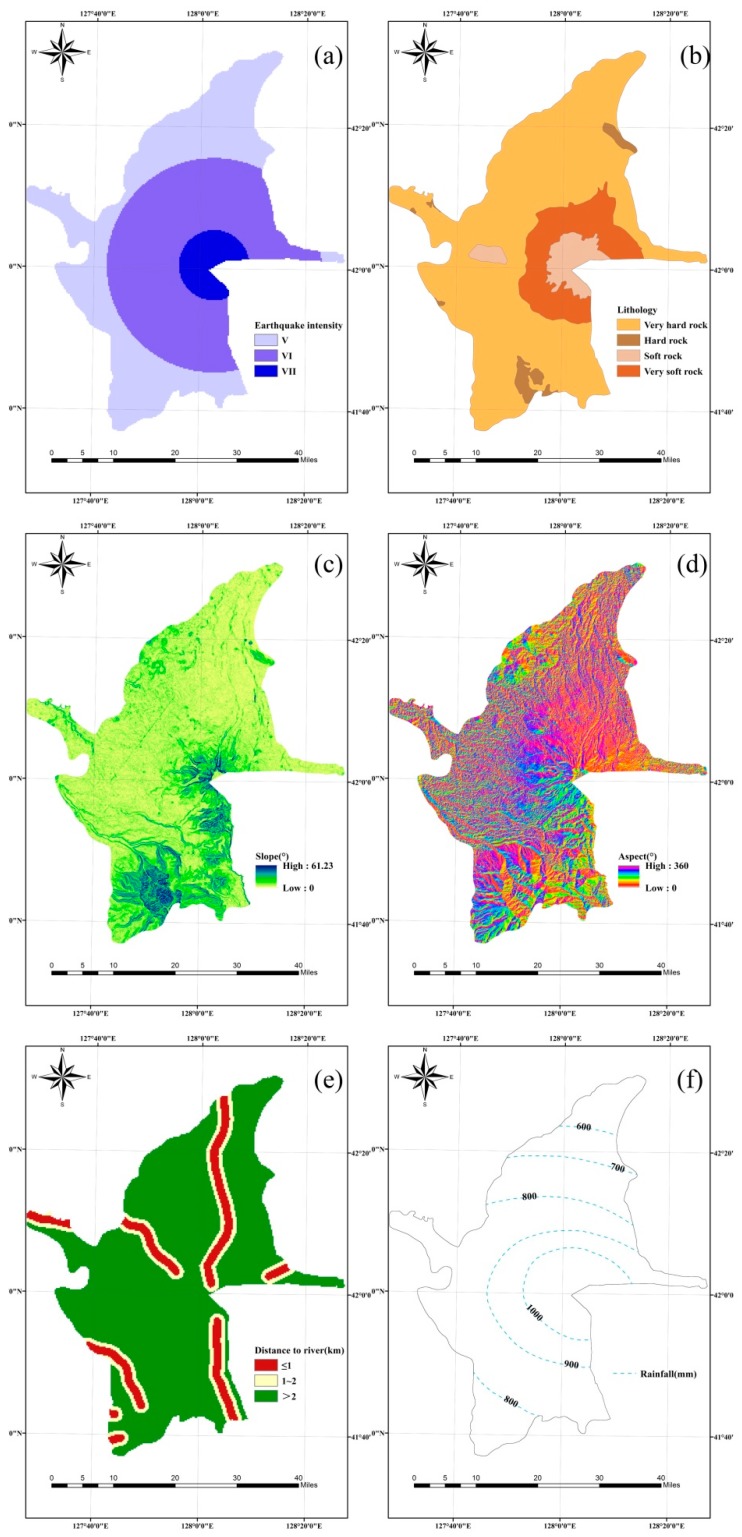
Condition parameters: (**a**) Earthquake intensity, (**b**) lithology, (**c**) slope, (**d**) aspect, (**e**) distance to the river, and (**f**) precipitation.

**Figure 5 ijerph-16-03330-f005:**
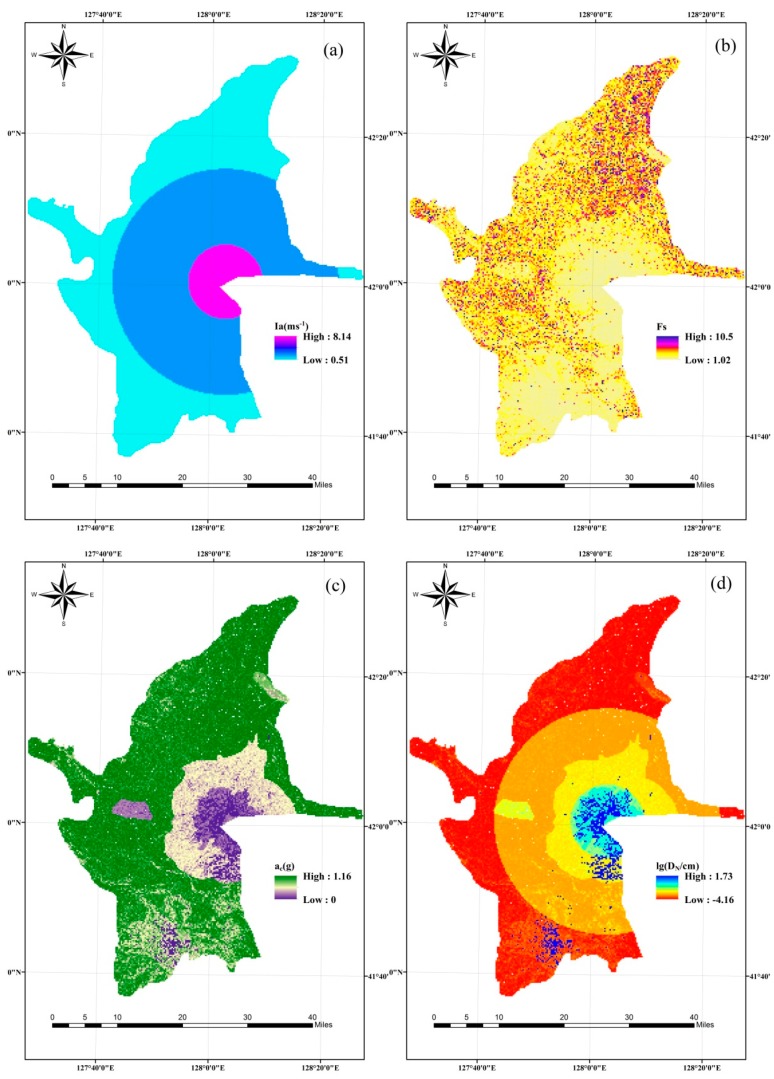
Condition parameters: (**a**) Arias intensity, (**b**) static safety factor, (**c**) critical acceleration, and (**d**) cumulative displacement.

**Figure 6 ijerph-16-03330-f006:**
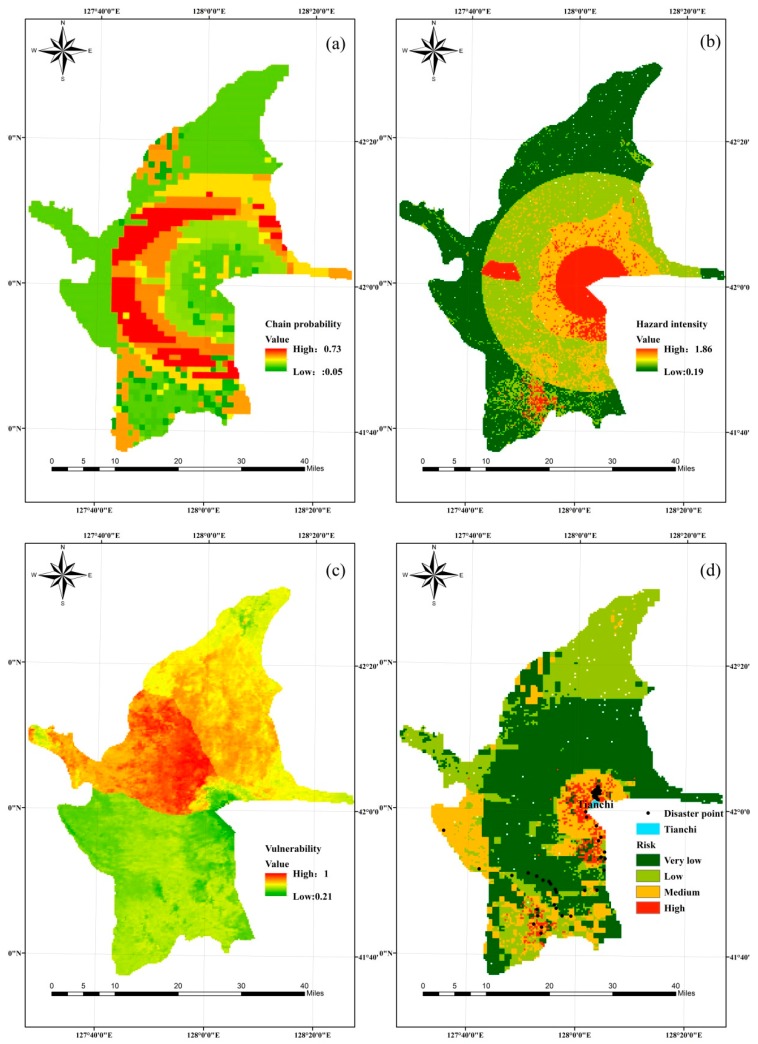
Earthquake–collapse–landslide disaster chain maps: (**a**) chain probability, (**b**) hazard intensity, (**c**) vulnerability, and (**d**) risk and disaster point locations.

**Figure 7 ijerph-16-03330-f007:**
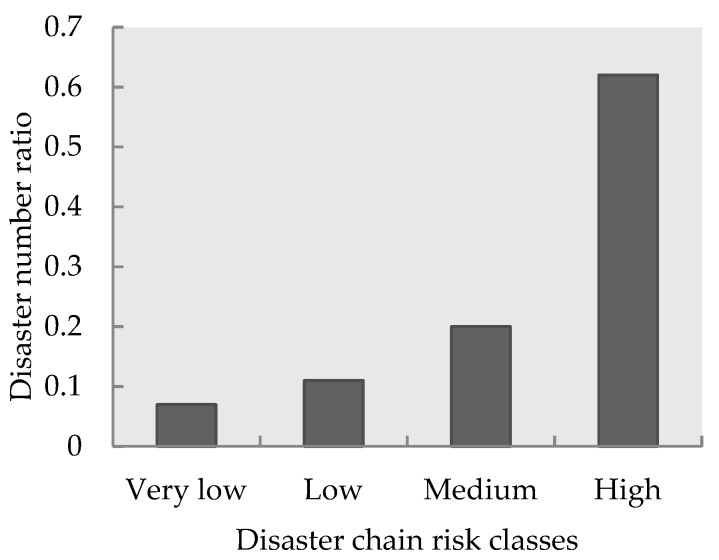
The relative distribution between the disaster chain risk zones and disaster number ratio.

**Figure 8 ijerph-16-03330-f008:**
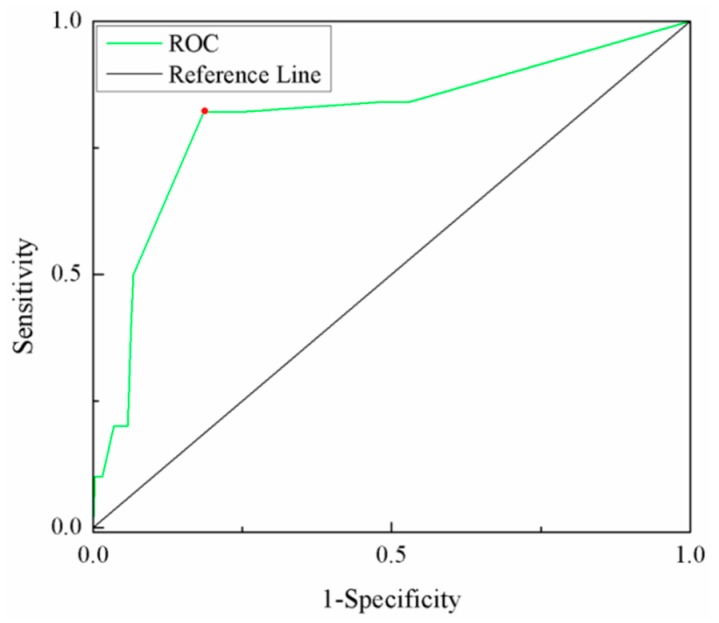
The receiver operating characteristic (ROC) curve of the risk results.

**Table 1 ijerph-16-03330-t001:** Categories of engineering rock mass grades and their parameter values (GB50218T-2014, China).

Engineering Geologic Types	$C^{\prime }\;(\mathrm{MPa})$	$\varphi \;{}^{\prime }(\text{°})$	$\gamma \;(\mathrm{kN}/m^{3})$
Hard layer group: basalt	>0.22	>37	>26.5
Secondary hard layer group: trachyte	0.12~0.22	29~37	>26.5
Secondary soft layer group: tuff	0.08~0.12	19~29	24.5~26.5
Soft layer group: mudstone	<0.08	<19	<24.5

**Table 2 ijerph-16-03330-t002:** The True Positives, True Negatives, False Positives, and False Negatives numbers of “best threshold” point.

Numbers	Risk			
Actual		occurrence	non-occurrence	summation
	occurrence	TP:41	FN:9	TP+FN:50
	non-occurrence	FP:1046	TN:4826	FP+TN:5872
	summation	TP+FP:1087	FN+TN:4835	5922

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
