# Peer review of "Risk Assessment of An Earthquake-Collapse-Landslide Disaster Chain by Bayesian Network and Newmark Models"

_ijerph, 2019, doi:10.3390/ijerph16183330_

Round 1

Reviewer 1 Report

This is a resubmitted manuscript that has been previously rejected by the reviewer. For the reference purpose, the reviewer's previous comments are attached here:

The reviewer must reject this manuscript for being considered to the journal. The reason for such decision making is attached:

The research methodology, approaches, results, discussions, and even the title are quite similar to one of the authors' previous work:

https://www.mdpi.com/2220-9964/8/5/210

Surprisingly, the authors did not discuss their previous work in this manuscript. This further brings concerns to the reviewer.

In the revised manuscript, the authors did not cite and discuss their previous work (https://www.mdpi.com/2220-9964/8/5/210). Hence, the reviewer must reject the revised manuscript for being considered to the journal.

Reviewer 2 Report

Dear authors,

I have reviewed this article from its initial submission till now. I must say that you have addressed all my comments and I believe that it is now ready to be published. If it is possible, I would still suggest to have an English speaking colleague of yours to proof read it.

Reviewer 3 Report

The part of your work concerning validation must be improved.

Now it is clear (from table 2) that your model can compute a quantity T_0^a (one value for each of the 5922 grid cells, is it?) in the range [-1,2]
(is it the true interval? why negative values are possible? please explain in the text).

But what you show in table 2 is not a classification table (or, better, a contingency table), but only the coordinates of three of the points of the ROC curve.

You should increase the number of the points (varying T_0^a from -1 to 2 in step of 0.1, for instance) in order to draw more precisely the ROC curve. For each value of T_0^a, you should compute the numbers of TP, TN, FP and FN (True Positives, True Negatives, False Positives, False Negatives)
and evaluate the threshold efficiency (for instance evaluating the quantity Eff=(TP+TN)/(TP+FP+TN+FN) and taking the value of T_0^a at which Eff attains its maximum value). For the chosen value of T_0^a, you should show in a table the values of TP, TN, FP and FN, whose sum should be 5922.

Another important remark concern the relationship of this your work with other your works published in this field, like the one in "International Journal of Geo-Information", doi:10.3390/ijgi8050210
Please, after citing it, clearly state in your new paper which are the improvements with respect to your previous work(s) on this subject.

Round 2

Reviewer 1 Report

The reviewer's concern in the first round review has not been fully addressed by the authors.

Regarding the relation between the current work and the authors' previous work (https://www.mdpi.com/2220-9964/8/5/210), the revisions offered by the authors (i.e. Line 92 to 101) is sloppy and not convincing. The reviewer can not see enough merit of this manuscript.

Reviewer 3 Report

Some improvements concerning the typographic/linguistic aspects of the work.

Page 4 line 145 (formula (1)): check the size and the fonts of the character P, keeping them uniform with respect to the ones used in the text.

Page 4 line 156: instead of "training and verification parts, among" write "training and test subsets. Among".

Page 5 line 190 (formula (4)): Put the argument of the logarithm in brackets (t_d).

Page 9 figure 4: Make sure the figure and the caption enter the same page. If necessary, make the figure smaller.

Page 13 line 362: "10" instead of "Ten"

Some remarks concerning statistical analysis.

Page 13, line 368: the criterion adopted to choose the "best thereshold" should be clearly stated in the paper. It seems that the point on the ROC curve at minimum distance from (0,1) has been chosen, but this is just my guess. If so, this is a valid criterion, but you should clearly state it in the paper. If not, write the criterion you have decided to choose.

Page 14, figure 8: please, put on the ROC curve (in a different color) the point corresponding to the "best threshold".
In the former version of your work, you stated that there was a parameter T_0^a, the variation of which allowed to choose the "best threshold". Please state (either in the caption or somewhere in the text) the range in which this parameter has been varied and the value for which the "best threshold" can be obtained.

Page 15, table 2: please note that the a-posteriory threshold efficiency is 82%, a value reasonably in agrement both with the AUROC and with the value obtained in the validation procedure on the test subset (page 13, line 363).

This should be clearly stated somewhere, for instance in the conclusions (only as a suggestion, page 15 after line 412).

Author Response

This manuscript is a resubmission of an earlier submission. The following is a list of the peer review reports and author responses from that submission.

Round 1

Reviewer 1 Report

This  paper  deals with a very interesting subject. Unfortunately, it is written in somewhat incomprehensible English.

I think that the Authors should seriously make a big effort to improve the quality of their work.

In all formulas  the number labelling the equation is too close to the mathematical expression. This position can confuse the reader. I think that you should format correctly each equation shifting the number to the right.

Page 4 line 143: "; P(A), P(B) are the marginal probabilities." instead of "; P(A), P(B) is marginal probability."

Page 5 line 167 equation (2): is lg the Briggs logarithm (base 10)? Please explain. Please, put the argument D_N into parenthesis. What does the +-0.656 symbol means? Is is an uncertainty range for the cumulative displacement? Please explain this point in the paper.

Page 5 line 175: please use for Ia the same notation used in line 167 (a in subscript, both I and a in italic font).

Page 5 line 178: "where M is the eathquake magnitude according to Richter scale" instead of "where M is the Richter scale"

Page 5 line 181 equation (5):  r is a symbol and should be typed using italic fonts. ln the natural logarithm (base e)? Please explain in the paper.

Page 9 Figure 5 (d) What does it means that the logarithm of the cumulative displacement can reach 8.06? A Displacement of 10^8 cm seems to be unbelievable!!!!

Page 10 Figure 6 You should put the Units on the colormaps. Does "Chain probability Low-High" e means that probability ranges from 0 (Low) to 1 (High)?

For the other quantities (Hazard Intensity and Vulnerability), the ranges are not so evident!! As far as panel (d), you should explain somewhere (either in the caption or in the text) which variable was used for classification and the threshold used to break the range of this variable in four intervals corresponding to the four Risk levels.

Page 11 lines 320 and following: Which is the criterion used to assign each disaster chain risk to its category? This is not an on/off choice, you should have a classification criterion which should be clearly stated in the paper.

Page 11 lines 322 The method used in your work for model validation seems to be not compliant to the ones usually presented in the literature.

Usually, the dataset is partitioned into a training set used to build the model, and a test set used to check model accuracy. In your work, it seems that the same dataset is used both for model construction and for the estimate of its prediction capability. Please assess in the correct way the predictive capability of your model.

Page 12 line 356 figure 8: please use same unit on both axes, so to obtain a square box (same scale from 0 to 1 on both axis)

Usually, ROC curves obtained elaborating a set of "real" data are quite rough. Why your curve is so smooth? Please add information about how ther ROC curve has been computed. 

Which is the "best threshold" in order to have the highest prediction efficiency (also for a simpler prediction like "Risk/No risk")?

Is it possible to insert a contingency table to show the "best" classification obtainable?

Reviewer 2 Report

This is an improved version of the original manuscript that was submitted in Sustainability journal. Although the authors have made notable changes, they need to be more consistent on addressing the most important reviewer's comments, even if they disagree (in which case they can express their opinion in the response-to-reviewers letter). That said, I would suggest for this manuscript not to be published unless the authors proceed to some major improvements both in terms of text structure and methodology.

First of all, an English speaking colleague should check and rephrase the whole text before the authors resubmit it. During the first review I spent hours to try and fully understand the meaning of the manuscript, made numerous suggestions for improvements but the revised paper is still not suitable for publication.

The authors should further improve their paper by addressing the following comments:

1) Line 137: The cited literature deals with totally different phenomena compared to the recurrence period of earthquakes and especially strong earthquake events that can cause disaster chains. It is very difficult to come up with the probability of earthquake occurrence and most importantly, this can't be estimated using historical data. Please explain how this methodology is suitable for the earthquakes.

2) Provide technical details regarding your work in ArcGIS. What's the cell size of your DEM? How is probability incorporated in the maps of Figure 6? How did you produce those maps?

3) Throughout the text there is an extended and repetitive theoretical presentation about conditional probabilities but the actual probability values used are not supported anywhere in the text. This is one of the most critical parts of this methodology and it is not discussed at all.

4) The manuscript fails to present the scientific background and relevant assumptions regarding the use of BN and Newmark methods. Apart from that, I'm afraid that the use of these methods in order to assess seismic hazard and related catastrophic phenomena is not suitable. There is a huge scientific background and a vast amount of methodologies on seismic hazard and landslide hazard assessment and the authors should try and put their methodology in this context.

5) The authors do not refer to any methodology regarding the actual probability of earthquake recurrence or to other hazards (For example, there is no explanation on how the authors calculated Landslide susceptibility. Which factors did they take into account?). Apart from that, I have my doubts about the clarity and validity of this method, as it is created only for one event, based on that same event and back tested against that same event.

Reviewer 3 Report

The reviewer must reject this manuscript for being considered to the journal. The reason for such decision making is attached:

The research methodology, approaches, results, discussions, and even the title are quite similar to one of the authors' previous work:

https://www.mdpi.com/2220-9964/8/5/210

Surprisingly, the authors did not discuss their previous work in this manuscript. This further brings concerns to the reviewer.